# Effects of Extraction Methods on Phenolic Content in the Young Bamboo Culm Extracts of *Bambusa beecheyana* Munro

**DOI:** 10.3390/molecules27072359

**Published:** 2022-04-06

**Authors:** Mohd. Izuddin Nuzul, Vivien Yi Mian Jong, Lee Feng Koo, Thye Huat Chan, Chung Huap Ang, Juferi Idris, Rafidah Husen, Siaw Wei Wong

**Affiliations:** 1Centre of Applied Science Studies, Universiti Technologi MARA, Kota Samarahan 94300, Sarawak, Malaysia; mohdizuddinnuzul95@gmail.com (M.I.N.); angch@uitm.edu.my (C.H.A.); rafidahh@uitm.edu.my (R.H.); 2Department of Basic Sciences and Engineering, Faculty of Agriculture and Food Science, Universiti Putra Malaysia, Bintulu Campus, Bintulu 97008, Sarawak, Malaysia; leefeng@upm.edu.my; 3Carbon Xchange (Sarawak) Sdn. Bhd. 1st Floor, Lot 8724, Block 16, 17-C, Green Heights PH3, New Airport Road, Kuching 93250, Sarawak, Malaysia; bambusa@yahoo.com (T.H.C.); jesswsw@gmail.com (S.W.W.); 4Faculty of Chemical Engineering, College of Engineering, Universiti Teknologi MARA (UiTM), Sarawak Branch, Samarahan Campus, Kota Samarahan 94300, Sarawak, Malaysia; juferi@uitm.edu.my; 5Faculty of Chemical Engineering, College of Engineering, Universiti Teknologi MARA (UiTM), Shah Alam 40450, Selangor, Malaysia

**Keywords:** *Bambusa beecheyana*, total phenolic content, total flavonoid content, free radical scavenging activity, cinnamic acid derivatives

## Abstract

Nowadays, many studies focus on the potential of bamboo as a source of bioactive compounds and natural antioxidants for nutraceutical, pharmaceutical, and food sources. This study is a pioneering effort to determine the total phenolic content, total flavonoid content and free radical scavenging activity, as well as the phenolic identification and quantification of *Bambusa beecheyana*. The study was conducted by using ethanol, methanol, and water for solvent extraction by applying cold maceration, Soxhlet, and ultrasonic-assisted extraction techniques. The results showed that Soxhlet and ultrasonic-assisted *Bambusa beecheyana* culm extracts had an increase in the extract’s dry yield (1.13–8.81%) but a constant *p*-coumaric acid (**4**) content (0.00035 mg/g) as compared to the extracts from the cold maceration. The ultrasonic-assisted extraction method required only a small amount (250 mL) of solvent to extract the bamboo culms. A significant amount of total phenolics (107.65 ± 0.01 mg GAE/g) and flavonoids (43.89 ± 0.05 mg QE/g) were found in the Soxhlet methanol culm extract. The extract also possessed the most potent antioxidant activity with an IC_50_ value of 40.43 µg/mL as compared to the positive control, ascorbic acid. The UHPLC–ESI–MS/MS analysis was carried out on the Soxhlet methanol extract, ultrasonic-assisted extract at 40 min, and cold methanol extract. The analysis resulted in the putative identification of a total of five phenolics containing cinnamic acid derivatives. The two cinnamic acid derivatives, *p*-coumaric acid (**4**) and 4-methoxycinnamic acid (**5**), were then used as markers to quantify the concentration of both compounds in all the extracts. Both compounds were not found in the water extracts. These results revealed that the extract from Soxhlet methanol of *Bambusa beecheyana* could be a potential botanical source of natural antioxidants. This study provides an important chemical composition database for further preclinical research on *Bambusa beecheyana*.

## 1. Introduction

Similar to corn, wheat, rye, oat, sugarcane, barley, and rice, bamboo is a grass that belongs to the Poaceae family, where it covers over 1250 species from 75 genera. It is normally found within tropical, subtropical, and temperate parts of all countries [1]. In addition, with a total area of over 10 million hectares in southern Asia and approximately 5 million hectares in China, Asia is the largest bamboo reservoir in the world [2]. In Asia, bamboo is a common main ingredient for vegetable dishes, salads, and pickles. Bamboo has been described as a great supply of nutrients, minerals, amino acids, and dietary fiber. These contain phenolic compounds which contribute to their significant antioxidant potential [3,4,5,6,7,8]. However, the scientific study on the chemical composition and potential application of *Bambusa beecheyana* species is still very limited. Therefore, various pieces of research need to be done to determine the potential of this species as a nutraceutical and food source.

Given the wide range of bioactive constituents that might present in many plant species, a standard and integrated approach to extracting these potential bioactive components is required. Hence, selecting an appropriate extraction method is important in producing potential bioactive compounds that are rich in phenolics, especially bioactive flavonoids. Conventional extraction methods such as maceration and Soxhlet are common methods used to produce botanical extracts, while modern extraction methods such as ultrasonic-assisted extraction (UAE) have extensively been used in the optimization of phenolic extraction. Both maceration and Soxhlet required a longer time and more solvents for the extraction process compared to UAE. However, conventional methods have a greater advantage in terms of extract yield compared to UAE due to the large amount of sample used in the extraction process [9,10]. Various extraction techniques were used in the production of botanical extracts depending on the sample type, desired bioactivity, and targeted and non-targeted metabolites.

Maceration is a technique where the plant materials are soaked in a solvent and placed in a closed container for at least three days [11]. Ethanol, methanol, and water are commonly used solvents for the extraction of *Bambusa* species. Bioactive compounds such as phenolic, flavonoid, alkaloid, terpenoid, and tannin were found in various studies on *Bambusa* species [12,13,14,15,16]. Meanwhile, Soxhlet is a technique used widely for the extraction of thermally stable compounds. It is a continuous cycle extraction through the matrix by boiling and condensation and the sample in collected in the hot solvent [17]. Phenolic and flavonoid compounds were mainly found in the previous study [12,18]. Moreover, ultrasonic-assisted extraction uses ultrasonic waves to disrupt the plant matrix in order to accommodate the release of bioactive compounds [19]. Likewise, phenolic and flavonoid compounds were found in the extract [20]. However, there was a limited study on the use of ultrasonic-assisted to extract the plant material from *Bambusa* species. All the extraction methods have shown to be good extraction methods used to extract phenolic and flavonoid compounds; thus, they have a promising antioxidant activity that are contributed from those compounds. Methanol is a commonly used solvent even though it is identified as a toxic solvent. Due to its high polarity, methanol influences the extract yields and resulting antioxidant activities [21].

In the present study, different extraction methods and solvents have been used to determine the total phenolic and flavonoid content as well as antioxidant activity of *Bambusa beecheyana* culm extracts. The effect of the different extraction techniques on the extraction yield, total phenolic content, total flavonoid content, and the antioxidant activity of *Bambusa beecheyana* culm extracts was revealed for the first time in this study. UHPLC-ESI-QTOF-MS/MS analysis was conducted to identify the potential biomarkers. Moreover, in-depth bioactive compounds were characterized and isolated using HPLC, GC/MS, UV-Vis, FTIR, and NMR.

## 2. Results

### 2.1. Extraction of Bambusa beecheyana Culms

*Bambusa beecheyana* culms were first extracted with ethanol, methanol, and water using cold maceration to measure its extraction efficiency. *p*-coumaric acid and 4-methoxycinnamic acid are known as cinnamic acid derivatives. Both derivatives were known to be strong antioxidants [22]. Therefore, the determination of both compounds was significant in this experiment. However, there was no significant change in dry yield (1.06–2.00%) for *p*-coumaric acid (**4**) (0.00039 to 0.00059 mg/g), but a noticeable change in 4-methoxycinnamic acid (**5**) content (0.00093–0.00278 mg/g) after cold maceration extraction (see Table 1). After this, the *Bambusa beecheyana* culms were extracted using Soxhlet and ultrasonic-assisted extraction (UAE) with the same solvents, which were methanol, ethanol, and water, respectively (see Table 2). This increased the extract’s dry yield (1.13–8.81%), but had the same trend in *p*-coumaric acid (**4**), and 4-methoxycinnamic acid (**5**) content. The *p*-coumaric acid (**4**) showed a constant content of 0.00035 mg/g.

The percentage of *p*-coumaric acid (**4**) was consistent with the respective extracts, although the percentage of dry yield varied among the extracts. Furthermore, the extraction methods and solvents used in these studies do not affect the percentage of the *p*-coumaric acid (**4**). The lower content of the *p*-coumaric acid (**4**) is affected by the lignification in the culms of the bamboo. In this study, the young bamboo culm was used; the consistent and lower content of *p*-coumaric acid (**4**) is due to lignification at the early stage of the development of the culm. The content of *p*-coumaric acid (**4**) will increase eventually at the later stage when the bamboo matures [23]. From the results, both *p*-coumaric acid (**4**) and 4-methoxycinnamic acid (**5**) were not detected in the water extracts for all three extraction techniques but were only found in the organic solvents of different polarities. To our advantage, the UAE method required only a small amount (250 mL) of solvent to extract the bamboo culms.

### 2.2. Isolation and Purification of p-Coumaric Acid (**4**), and 4-Methoxycinnamic Acid (**5**)

Two cinnamic acid derivatives, *p*-coumaric acid (**4**), and 4-methoxycinnamic acid (**5**) were isolated from the cold maceration ethanol extract (BBER) and the HPLC chromatogram as shown in Figure 1, with the retention time of 6.825 and 6.665, respectively. The HPLC chromatograms of both *p*-coumaric acid and 4-methoxycinnamic acid are shown in (Appendix A). Both compounds were separated using reversed-phase prep-HPLC. BBER was chosen because the peaks for both compounds were more significant than other extracts.

### 2.3. NMR Analysis of p-Coumaric Acid (**4**) and 4-Methoxycinnamic Acid (**5**)

The ^1^H-NMR spectrum of *p*-coumaric acid (**4**) [24,25] established the presence of four aromatic protons at δ 7.56 (2H, d, *J =* 8.64) and δ 6.91 (2H, d, *J =* 8.60). This showed the typical pattern for para-substituted aromatic moiety. Another two protons were assigned to the trans-olefinic protons at δ 7.61 (1H, d, *J* = 16.00) and δ 6.35 (1H, d, *J =* 16.00). According to its ^13^C-NMR spectrum, downfield signals at δ 174.0, δ 157.7, δ 144.3, and δ 130.0 were assigned carbon of the aromatic ring. The three high-field signals appearing at δ 126.2, δ 115.8, and δ 114.8 were assigned to the carbon atoms bearing carboxylic groups.

The ^1^H-NMR spectrum of 4-methoxycinnamic acid (**5**) [26] established the presence of four aromatic protons at δ 7.55 (2H, d, *J* = 8.56) and δ 6.91 (2H, d, *J* = 8.56). This showed the typical pattern for para-substituted aromatic moiety. Another two protons were assigned to the trans-olefinic proton at δ 7.61 (1H, d, *J* = 15.96) and δ 6.35 (1H, d, *J* = 15.96). A single sharp peak at δ 3.73 ppm was assigned to the methoxy. According to its ^13^C-NMR spectrum, a high-field signal at δ 50.6 was indicative for the presence of methoxy. Downfield signals at δ 167.4, δ 160.1, δ 144.6, and δ 130.0 were assigned carbon of the aromatic ring. The three high-field signals appearing at δ 125.8, δ 115.8, and δ 114.2 were assigned to the carbon atoms bearing carboxylic groups.

### 2.4. Contents of Bioactive Markers, p-Coumaric Acid (**4**), and 4-Methoxycinnamic Acid (**5**) in the Exctracts of Bambusa beecheyana

The results of regression analysis on calibration curves and detection limits are presented in Table 2. The detection limit was evaluated based on a signal-to-noise ratio of 3 (S/N = 3), the detection limit was 0.002 mg/mL. The quantitation limit was evaluated based on a signal-to-noise ratio of 10 (S/N = 10), the quantitation limit was 0.006 mg/mL.

The samples were injected directly and separated under the optimum condition. The chromatogram of one of the extracts is shown in Figure 1. The HPLC chromatograms of other extracts are shown in (Appendix A). The calculated content of 4- *p*-coumaric acid (**4**) and 4-methoxycinnamic acid (**5**) are shown in Figure 2. From the results, BBER has the highest content of compounds (**4**) and (**5**), which were 0.00059 mg/g ± 1.67 x 10^−11^ and 0.00278 mg/g ± 1.67 × 10^−11^, respectively. BBES, BBMS, BBEU60, BBMU40, and BBMU60 have the lowest content of compound (**4**) which is 0.00035 mg/g ± 1.67 × 10^−11^, 0.00035 mg/g ± 1.67 × 10^−11^, 0.00035 mg/g ± 1.67 × 10^−11^, 0.00035 mg/g ± 1.67 × 10^−11^, and 0.00035 mg/g ± 1.67 × 10^−11^ while BBEU20 has the lowest content of compound (**5**) which is 0.00002 mg/g ± 6.67 × 10^−11^. The uncertainty of the results was measured as %RSD with a confidence interval (95%) of 0.00008 for *p*-coumaric acid and 0.0006 for 4-methoxycinnamic acid. Both compounds were not detected in water extracts.

### 2.5. Total Phenolic Content (TPC)

The results were derived from a calibration curve (y = 0.0004x − 0.0163, R^2^ = 0.9996) of gallic acid (200–1000 µg/mL) and expressed in gallic acid equivalents (GAE) per gram of weight of dry extracts. From Table 3, BBMS has the highest total phenolic content which is 107.65 mg GAE/g, while 60 min BBHU60 has the lowest total phenolic content which is 27.89 mg GAE/g. From the results, methanolic extracts have the most phenolic compounds as compared to other extracts.

### 2.6. Total Flavonoid Content (TFC)

The results were derived from a calibration curve (y = 0.0197x − 0.0889, R^2^ = 0.9964) of quercetin (25–125 µg/mL) and expressed in quercetin equivalents (QE) per gram of weight of dry extracts. From Table 3, BBMS has the highest total flavonoid content which is 48.89 mg QE/g, while BBHR has the lowest total flavonoid content which is 12.38 mg QE/g. The results were the same trend as TPC where methanolic extracts have the highest TFC. From the results, it is shown that the TFC is significantly lower than the TPC, from which it can be suggested that flavonoid compounds were not the major class of phenolic constituents in *Bambusa beecheyana* extracts.

### 2.7. 2,2-Diphenyl-1-picrylhydrazyl (DPPH) Radical Scavenging Activity

DPPH is a stable free radical compound and has a strong absorption at 510 nm. DPPH free radical scavenging assay is a relatively rapid and efficient method to evaluate free radical scavenging activity [27]. The decreases in absorbance of DPPH radical can be observed by changes in color from purple to yellow. This shows that there is an interaction between the antioxidants present in the extracts and free radicals from DPPH [28]. DPPH free radical scavenging assay IC_50_ value can be determined as follows: 4250 µg/mL, inactive; 4100–250 µg/mL, weakly active; 450–100 µg/mL, moderately active; 10–50 µg/mL, strongly active; <10 µg/mL, very strongly active [29]. Based on the results shown in Table 3, BBMS exhibits moderate antioxidant activity, while BBER, BBMR, BBES, BBMU20, BBMU40, BBMU60 exhibit weak antioxidant activity.

### 2.8. Potential Bioactive Markers from Bambusa beecheyana Extracts

Liquid chromatography coupled with mass spectrometry (LC-MS) has been widely used to identify organic compounds, where it is more sensitive and accurate. Three extracts from each extraction method that possess the highest total phenolic and flavonoid content and DPPH radical scavenging activity were chosen to be further analyzed by using UHPLC-ESI-QTOF-MS/MS to identify the potential bioactive markers. The compounds were identified by utilizing their MS/MS spectra from the library and comparison with literature data. Table 4 shows a list of potential bioactive chemical markers found in those three extracts. Chromatograms of the three extracts are shown in Figure 3.

From the analysis, a total of five phenolic compounds were putatively identified based on the MS/MS data in comparison with the literature. The base peak chromatograms showed that most of the prominent peaks were attributed to the presence of those phenolic compounds. Ferulic acid (**1**), cinnamic acid (**2**), *p*-coumaric acid (**4**), and 4-methoxycinnamic acid (**5**) were identified as cinnamic acid and its derivatives. They have pseudomolecular ion peaks at *m*/*z* 193.0624, *m*/*z* 147.0585, *m*/*z* 163.0325, and *m*/*z* 177.0480, respectively. The fragment ion at *m*/*z* 146.0504 is due to the loss of hydroxyl and methoxy groups (Figure 4). This was proven by previous research done by [30,31,32]. Compound (**3**) was identified as hydroxybenzoic acid with pseudomolecular ion peak at *m*/*z* 137.0179 based on a comparison with a previous report [30]. This could support the high TPC and TFC values, and hence the potent antioxidant activities of the three *Bambusa beecheyana* extracts. The MS fragmentation of ferulic acid (**1**), *p*-coumaric acid (**4**) and 4-methoxycinnamic acid (**5**) matched the biosynthetic pathways (see Figure 5) of cinnamic acid, where it was part of the phenylpropanoid pathway. Phenyl ammonia lyase (PAL) converted phenylalanine to cinnamic acid. Then, cinnamic acid-4-hydroxylase hydroxylated the cinnamic acid to yield *p*-coumaric acid (**4**) [33]. From the pathways, it was clearly shown that ferulic acid (**1**), *p*-coumaric acid (**4**) and 4-methoxycinnmaic acid (**5**) were all the products of the cinnamic acid biosynthetic pathway. This could support the substantial amount of TPC and TFC values, and hence the potent antioxidant activities of the three *Bambusa beecheyana* extracts.

## 3. Discussions

Extraction is a crucial part of extracting the bioactive compounds from plants. Conventional and modern extraction methods have been widely used in the industry to achieve common goals which are: first, extracting targeted bioactive compounds from a complex plant sample; second, increasing the selectivity of the analytical method; third, increasing bioassay sensitivity; fourth, converting bioactive compounds into a more suitable form for detection and separation; and finally, providing a strong and repeatable method that is independent of variations in the sample matrix [34]. This is the pioneering study on different extraction methods that were applied to extract *Bambusa beecheyana* culms. 

The dry yield extracts were mainly affected by the time and solvent used in each extraction. Cold maceration and Soxhlet required a longer time and cold maceration required a larger volume of solvents. Meanwhile, UAE required a shorter amount of time and a smaller volume of solvents for extracting plant materials. In the present study, cold maceration was first used as the extraction method to obtain the bamboo culm extracts. No significant differences were observed in dry yield and *p*-coumaric acid (**4**), but there was a noticeable change in 4-methoxycinnamic acid (**5**) content after cold maceration extraction. The bamboo culms were extracted using Soxhlet, and ultrasonic-assisted extraction (UAE) with the same solvents increased the extract’s dry yield (1.13–8.81%), but the same trend in *p*-coumaric acid (**4**) and 4-methoxycinnamic acid (**5**) content. However, the dry yield percentage of UAE extracts is varied, whereas water extracts have the highest dry yield and ethanol extracts have the lowest dry yield. The same trend was reported by [35]. More polar solvents were expected to extract a higher number of hydrophilic compounds, thus resulting in a higher dry yield. 

In addition, the TPC and TFC of *Bambusa beecheyana* extracts were determined in this study. Methanol extracts possessed the highest amount of both TPC and TFC, while water extracts possessed the lowest TPC and TFC. Despite methanol being known as a toxic solvent, it is a widely used solvent due to its ability to extract different types of chemical constituents and produce higher yields of plant extract. A previous study reported that possibly the content of water extract was mainly non-phenol compounds such as carbohydrates and terpene [36]. The results (see Appendix A) have shown a positive correlation between the TPC and scavenging activity of the extracts (R^2^ = 0.3294) with *p* < 0.05, meanwhile the correlation between TFC and scavenging activity of the extracts (R^2^ = 0.4538) also showed a positive correlation with *p* < 0.05. In addition, higher TPC is attributed to the potent antioxidant activity which has been reported in a previous study [37]. Furthermore, the antioxidant activity of the phenolic compounds was also influenced by their structural composition. This is mainly due to the capacity of phenolics to scavenge free radicals [38]. The antioxidant activity of phenolic compounds may also be related to the electron-donating ability of the carboxylic acid group. In addition, the presence of hydroxy and methoxy groups in the phenolic compounds will also enhance the free radical scavenging activity based on the number and position of both groups [39]. The ortho position in an aromatic ring paired with the presence of di-active groups, especially with the presence of the hydroxyl group, will possess high free radical scavenging activity due to the hydroxyl group having lower bond dissociation energies [40]. This trend can be seen in these extracts: BBER, BBMR, BBES, BBMS, and BBMU40 showed a potent DPPH scavenging activity with IC_50_ values of 95.93 ± 0.02 µg/mL, 63.32 ± 0.04 µg/mL, 87.12 ± 0.03 µg/mL, 40.43 ± 0.02 µg/mL, and 45.01 ± 0.03 µg/mL, respectively. However, secondary metabolites such as vitamin C and E and traces of mineral elements such as selenium, copper, zinc, manganese, and iron also contributed to the antioxidant activity of bamboo extract [3].

Table 5 shows the comparison of TPC, TFC, and DPPH scavenging activity of *Bambusa beecheyana* and various species of bamboo. The values varied based on the species and demographic of the bamboo. The study of bamboo species is still limited. From the table, *Bambusa arundinacea* has the highest TPC and TFC among the *Bambusa* genus, which is 647.76 mg GAE/g and 247.85 mg QE/g, respectively. The *Bambusa* genus has great potential as a medicinal plant. It has promising antioxidant activity due to the phenolic and flavonoid content which is a source of antioxidants. Further research on the potential of the *Bambusa* genus as a great source of antioxidants can be done in the future.

Additionally, putative identification of potentially bioactive compounds was done by using UHPLC-ESI-QTOF-MS/MS. From the analysis, cinnamic acid and its derivatives were found in *Bambusa beecheyana* extract; three of its derivatives are *p*-coumaric acid (**4**), 4-methoxycinnamic acid (**5**), and ferulic acid (**1**). Two of the derivatives were successfully isolated from the extract: *p*-coumaric acid (**4**) and 4-methoxycinammic acid (**5**). Cinnamic acid is a phenolic compound which is known to be an antioxidant [43] and anticancer [44]. While its methoxylated derivatives are potentially hepatoprotective, antidiabetic, neuroprotective, and chemopreventive [45]. In this study, the contents *p*-coumaric acid (**4**) and 4-methoxycinnamic acid (**5**) isolated from the extract were not significantly high, whereas BBES had the highest content of both compounds which were 0.00059 mg/g ± 0.03 and 0.00278 mg/g ± 0.02, respectively; thus, the high TPC value and the antioxidant activity of the extracts were also contributed by other phenolic and secondary metabolites present in the extract. Therefore, further investigation can be done in the future to explore more medicinal potential of *Bambusa beecheyana* extracts.

## 4. Materials and Methods

### 4.1. Raw Material

The young culms of *Bambusa beecheyana* were collected from the plantation of Carbon Xchange Sdn. Bhd, Kuching. The plant authentication was carried out by Mr. Tinjan Anak Kuda from Jabatan Hutan and the herbarium voucher specimen (UiTM3040) was kept in UiTM Sarawak, Samarahan 2 Campus. The young culms of *Bambusa beecheyana* were dried under shade for 14 days, respectively, prior to mechanical grinding into a fine powder. The raw materials were stored in a chiller at 5 °C until needed. 

### 4.2. Extraction Method of Bambusa beecheyana Culm

500 g, 300 g, and 50 g of powdered young bamboo culm were weighed. Each sample was extracted using solvents which were deionized water, ethanol, and methanol. 

For cold extraction, the powdered bamboo (500 g) was soaked in ethanol (2 L) for 72 h [14]. Another 300 g of powdered young bamboo culm was weighed and extracted using Soxhlet extractor for six hours where 250 mL of solvent was used [46]. Meanwhile, using ultrasonic-assisted extract, the powdered young bamboo culm (50 g) was weighed and extracted using a bath sonicator and soaked in ethanol (2 L) for 20, 40, and 60 min. The water bath temperature was maintained at 60 °C. Then, all the extracts were filtered using vacuum filter. The extracts were then evaporated by using a rotary evaporator to separate the solvents from the samples. Finally, the crude extract was weighed. All the steps were repeated using methanol and deionized water.

### 4.3. Total Phenolic Content

The TPC was determined by using Folin–Ciocalteu method described by [15] with a slight modification. An amount of 5 mg of the sample was dissolved in 5 mL of ethanol. A total of 300 µL of the sample was taken out and placed into another vial. Then, 2250 µL of Folin–Ciocalteu reagent was added to it. The solution mixture was let stand for 5 min. An amount of 2250 µL of 6% of sodium carbonate was gently mixed into the vial. After 60 min, the absorbance values were measured by UV-Vis spectrophotometer (Lambda 25, Perkin Elmer, Waltham, MA, USA) with detection of 765 nm by using gallic acid (200, 400, 600, 800 and 1000 µg/mL) as the reference standard. The results were expressed as mg GAE/g extract.

### 4.4. Total Flavonoid Content

The TFC of the extracts was determined by using aluminum chloride colorimetric method as described by [47] with a slight modification. Briefly, 5 mL of sample was mixed with 5 mL of 2% aluminum chloride in a vial. The vial was shaken and left for 10 min. The analysis was carried out using UV-Vis spectrophotometer with the detection of 415 nm by using quercetin (25, 50, 75, 100 and 125 µg/mL) as the reference standard. The results were expressed as mg QE/g extract.

### 4.5. 2,2-Diphenyl-1-picrylhydrazyl (DPPH) Scavenging Activity Assay

A sample of 6 mg was dissolved in ethanol as a stock solution. It was diluted to concentrations of 50, 125, 250, 500, and 1000 µL/mL. An amount of 1 mL of 2,2-diphenyl-1-picrylhydrazyl (DPPH) was added into a vial containing a mixture of 1 mL of sample solutions at different concentrations and 3 mL of ethanol. The solution mixture was allowed to react for 60 min. The absorbance was measured at 517 nm using UV-Vis spectrophotometer with ethanol as blank, ascorbic acid as positive control, and 1 mL ethanol plus 3 mL DPPH as a negative control. The percentage of inhibition was calculated by using the following formula:% inhibition = [(Ab_c_ − Ab_s_)/Ab_c_] × 100%

Ab_c_ is the absorbance of negative control and Ab_s_ is the absorbance of samples.

This method was adapted from [27,48] with slight modifications. The results were expressed in IC_50_ value.

### 4.6. UHPLC-ESI-QTOF-MS/MS Analysis

The UHPLC-ESI-QTOF-MS/MS analysis was carried out using a 1290 Infinity UHPLC-ESI-QTOF-MS/MS system attached to a 6550 iFunnel Quadrupole Time of Flight mass spectrometer (Agilent, Santa Clara, CA, USA). Analyte separation was carried out using a Zorbax Eclipse Plus C18 column (ø 4.6 × 100 mm, 3.5 µm, Agilent, Santa Clara, CA, USA) with a mobile phase consisting of LCMS grade water (solvent A) and acetonitrile (solvent B), flowing at 1.0 mL/min. The programmed gradient consisted of 0 min (90% A), 10 min (0% A), and 15 min (0% A). The MS analysis was done with the parameters set as follows: negative and positive ion mode, fixed collision energies of 10, 20, 30, 40, and 50 eV, a gas temperature of 200 °C and a flow rate of 14 L/min, sheath gas temperature and flow rate of 350 °C and 11 (arbitrary units), respectively. Finally, the mass resolution was set to a full scan from 100 to 1700 amu. The identification of potential bioactive markers was done by comparing the data from MS/MS with the literature.

### 4.7. Isolation of Bioactive Compounds Using Preparative HPLC

The crude fraction was purified on a preparative HPLC (1260 Infinity, Agilent, Santa Clara, CA, USA) system with Zorbax Eclipse XDB-C18 (Agilent, Santa Clara, CA, USA) column (ø 9.4 mm × 250 mm; 5 µm). The mobile phase consisted of water (solvent A) and acetonitrile (solvent B). The gradient elution program was as follows: 0–10 min, 10% B; 10–15 min, 100% B. The flow rate was set at 10 mL/min with UV detection at 330 nm to yield compounds (**4**) (2.0 mg) and (**5**) (4.0 mg).

### 4.8. Quantification and Optimization of Compounds (**4**) and (**5**)

The analysis was done by HPLC (1260 Infinity Quaternary LC VL, Agilent, Santa Clara, CA, USA), with Zorbax Eclipse Plus C18 (Agilent, Santa Clara, CA, USA) column (ø 4.6 mm × 100 mm; 3.5 µm).

#### 4.8.1. Preparation of Standard Solutions

Standard stock solutions for compounds (**4**) and (**5**) were prepared in methanol at a concentration of 0.002, 0.004, 0.006, and 0.008 mg/mL. All the standard solutions were filtered using a membrane filter before being injected into HPLC.

The dried extracts (6 mg) were dissolved in methanol (6 mL). All the sample solutions were filtered using a 0.20 µm membrane filter before being injected into HPLC. The uncertainty of the results was measured in terms of %RSD.

#### 4.8.2. Chromatographic Conditions

The analysis was carried out by a C18 reversed-phase column. The mobile phase was water (Solvent A) and acetonitrile (Solvent B). Before using the mobile phase, it was filtered using a membrane filter and de-aerated ultrasonically. The gradient elution program was as follows: 0–10 min, 10% B; 10–15 min, 100% B. Compounds (**4**) and (**5**) were quantified by DAD following RP-HPLC separation at 330 nm. The flow rate and injection volumes were 1.0 mL/min and 10 µL, respectively. The peaks of the analytes were confirmed by comparing the retention time and UV spectra of the compounds isolated. Quantification was carried out by the integration of the peak using a standard method. The operations were carried out at ambient temperature. This method was adapted and modified from [49].

### 4.9. Analytical Data

The characterization of two cinnamic acid derivatives; *p*-coumaric acid (**4**) and 4-methoxycinnamic acid (**5**) were done as follows: determination of UV wavelength was carried out using UV-Vis spectrophotometer (Lambda 25, Perkin Elmer, Waltham, MA, USA). The mass spectrometry analysis was carried out using GC (Clarus 680, Perkin Elmer, Waltham, MA, USA) coupled with MS (Clarus Sq 8T, Perkin Elmer, Waltham, MA, USA). The characterization of functional groups was performed on FTIR-ATR (Frontier, Perkin Elmer, Waltham, MA, USA). NMR was carried out using Bruker 400 MHz (USA).

(*E*)-3-(4-hydroxyphenyl)prop-2-enoic acid (**4**) [24,25]. Yellowish solid. UV λ_max_ (MeOH) 330 nm; EIMS *m*/*z* 164.08; IR(ATR) ν_max_/cm 3347, 2925, 2853, 1631; ^1^H NMR (400 MHz, acetone-d_6_) 7.61 (d, *J* = 16.0 Hz, 1H), 7.56 (d, *J* = 8.64 Hz, 2H), 6.91 (d, *J* = 8.60 Hz, 2H), 6.35 (d, *J* = 16.0 Hz, 1H); ^13^C NMR (400 MHz, acetone-d_6_) 174.0 (C), 157.7 (C), 144.3 (CH), 130.0 (CH), 126.2 (C), 115.8 (CH), 114.8 (CH).

(*E*)-3-(4-methoxyphenyl)prop-2-enoic acid (**5**) [26]. Yellowish solid. UV λ_max_ (MeOH) 330 nm; EIMS *m*/*z* 178.02; IR(ATR) ν_max_/cm 3369, 2920, 2850, 1685, 1633, 1282; ^1^H NMR (400 MHz, acetone-d_6_) 7.61 (d, *J* = 15.9 Hz, 1H), 7.55 (d, *J* = 8.56 Hz, 2H), 6.91 (d, *J* = 8.56 Hz, 2H), 6.35 (d, *J* = 15.96 Hz, 1H), 3.73 (s, 3H); ^13^C NMR (400 MHz, acetone-d_6_) 167.4 (C), 160.1 (C), 144.6 (CH), 130.0 (CH), 125.8 (C), 115.8 (CH), 114.2 (CH), 50.6 (CH_3_).

### 4.10. Statistical Analysis

Microsoft Excel 365 was used for the analysis of TPC, TFC, and DPPH scavenging activity and quantification of compounds (**4**) and (**5**). The results were expressed as the mean ± standard deviation of three replicates. Two-way ANOVA followed by Tukey’s comparison method was used to determine the significant difference between the extraction methods and the solvents used. *p*-values less than 0.05 were considered significant.

## 5. Conclusions

Extraction technique and solvent selection are important steps in sample preparation for nutraceuticals and food sources. Soxhlet and ultrasonic-assisted *Bambusa beecheyana* culms extracts showed an increase in dry yield but a constant *p*-coumaric acid (**4**) content as compared to the extracts from the cold maceration. A substantial amount of total phenolics (107.65 ± 0.01 mg GAE/g) and flavonoids (43.89 ± 0.05 mg QE/g) were found in the Soxhlet methanol culm extract. The extract also possesses the most potent antioxidant activity with an IC_50_ value of 40.43 µg/mL as compared to the positive control, ascorbic acid. From the UHPLC-ESI-QTOF-MS/MS analysis, a total of five phenolics were putatively identified and this analysis has led us to identifying cinnamic acid derivatives, *p*-coumaric acid (**4**) and 4-methoxycinnamic acid (**5**), which later were isolated. They were then used as markers to quantify the concentration of both compounds in all the extracts. Both compounds were found in the organic extracts (methanol and ethanol) but not in the water extracts. This indicated that organic solvents have the ability to extract different types of phytochemicals and produce higher yields of plant extract. This study highlighted the usage of the different extraction techniques and suitable solvents in bamboo culm extraction, which could benefit the application of bamboo culm extracts in preclinical research to support their medicinal benefits.

## Figures and Tables

**Figure 1 molecules-27-02359-f001:**
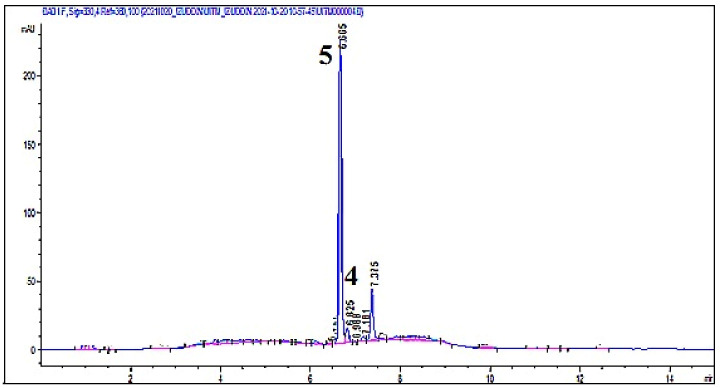
HPLC chromatogram of cold maceration ethanol extract (BBER).

**Figure 2 molecules-27-02359-f002:**
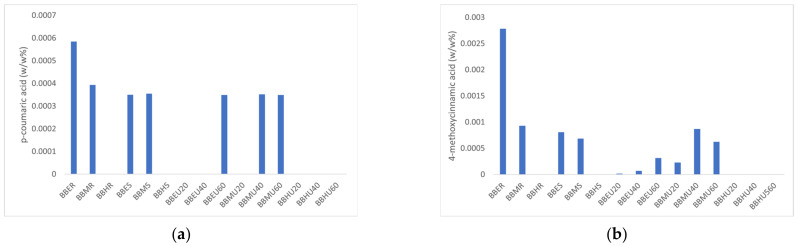
Content of compounds (**4**) and (**5**) in the extracts of *Bambusa beecheyana*. The results are reported in *w*/*w*%: (**a**) *p*-coumaric acid (**4**); (**b**) 4-methoxycinnamic acid (**5**).

**Figure 3 molecules-27-02359-f003:**
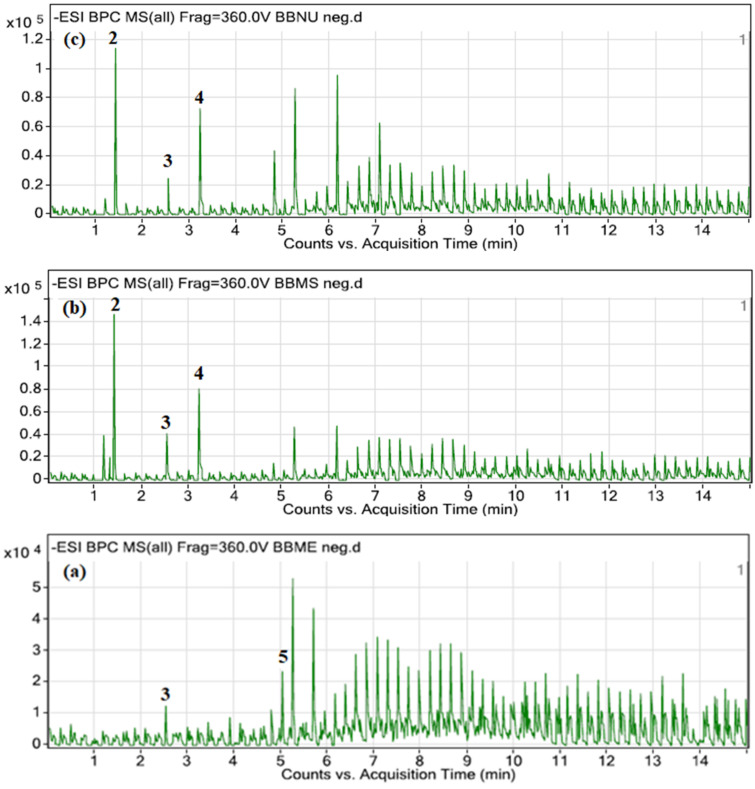
Total ion chromatogram (TIC) of: (**a**) BBMR, (**b**) BBMS, and (**c**) BBMU40.

**Figure 4 molecules-27-02359-f004:**
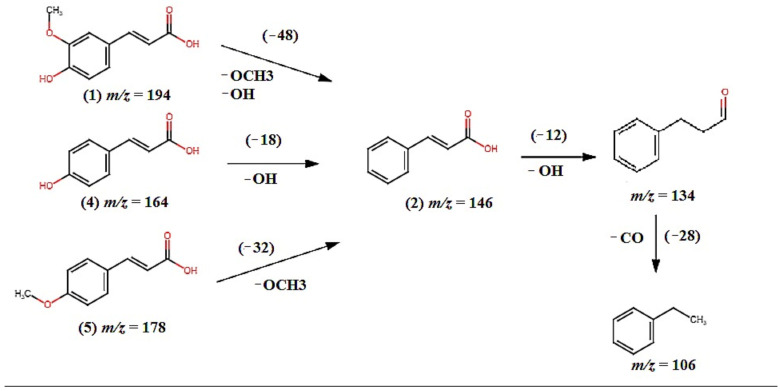
Fragmentation pathways for compound ferulic acid (**1**), cinnamic acid (**2**), *p*-coumaric acid (**4**), and 4-methoxycynammic acid (**5**).

**Figure 5 molecules-27-02359-f005:**
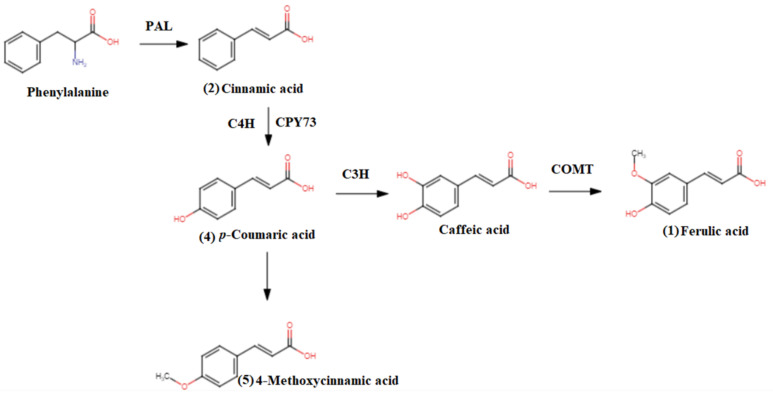
Biosynthetic pathways of cinnamic acid.

**Table 1 molecules-27-02359-t001:** Yield percentage of *Bambusa beecheyana* extracts.

Sample	Dry Yield (%)	*p*-Coumaric Acid (4) (mg/g)	%RSD	4-Methoxycinnamic Acid (5) (mg/g)	%RSD
Maceration					
BBER *	1.06 ± 0.0004	0.00059 ± 1.67 × 10^−11^	0.0000056	0.00278 ± 1.67 × 10^−11^	0.0000012
BBMR *	1.20 ± 0.0028	0.00039 ± 1.67 × 10^−11^	0.0000086	0.00093 ± 2.17 × 10^−10^	0.0000474
BBHR *	2.00 ± 0.0204	ND *	ND *	ND *	ND *
Soxhlet					
BBES *	4.12 ± 0.0010	0.00035 ± 1.67 × 10^−11^	0.0000096	0.00081 ± 1.17 × 10^−10^	0.0000283
BBMS *	5.09 ± 0.0011	0.00035 ± 6.67 × 10^−11^	0.0000039	0.00069 ± 2.17 × 10^−10^	0.0000065
BBHS *	4.35 ± 0.0007	ND *	ND *	ND *	ND *
Ultrasonic-assisted					
BBEU20 *	2.06 ± 0.0052	ND *	ND *	0.00002 ± 6.67 × 10^−11^	0.0005000
BBEU40 *	2.93 ± 0.0045	ND *	ND *	0.00007 ± 1.67 × 10^−11^	0.0000455
BBEU60 *	1.13 ± 0.0019	0.00035 ± 1.67 × 10^−11^	0.0000094	0.00032 ± 1.67 × 10^−11^	0.0000103
BBMU20 *	1.85 ± 0.0034	ND *	ND *	0.00023 ± 6.44× 10^−7^	0.1446180
BBMU40 *	2.84 ± 0.0027	0.00035 ± 1.67 × 10^−11^	0.0000094	0.00087 ± 2.17 × 10^−10^	0.0000502
BBMU60 *	2.42 ± 0.0014	0.00035 ± 1.67 × 10^−11^	0.00000096	0.00062 ± 2.17 × 10^−10^	0.0000681
BBHU20 *	6.64 ± 0.0035	ND *	ND *	ND *	ND *
BBHU40 *	8.81 ± 0.0025	ND *	ND *	ND *	ND *
BBHU60 *	7.77 ± 0.0011	ND *	ND *	ND *	ND *

* BBER—cold maceration ethanol extract, BBMR—cold maceration methanol extract, BBHR—cold maceration water extract; BBES—Soxhlet ethanol extract, BBMS—Soxhlet methanol extract, BBHS—Soxhlet water extract; BBEU20—20 min ultrasonic-assisted ethanol extract, BBEU40—40 min ultrasonic-assisted ethanol extract, BBEU60—60 min ultrasonic-assisted ethanol extract; BBMU20—20 min ultrasonic-assisted methanol extract, BBMU40—40 min ultrasonic-assisted methanol extract, BBMU60—60 min ultrasonic-assisted methanol extract; BBHU20—20 min ultrasonic-assisted water extract, BBHU40—40 min ultrasonic-assisted water extract, BBHU60—60 min ultrasonic-assisted water extract. ND—not detected.

**Table 2 molecules-27-02359-t002:** Validation data from calibration curves of compounds (**4**) and (**5**).

Compounds	Regression Equation	Correlation Coefficient (R^2^)	Linear Range (mg/mL)	Detection Limit (mg/mL)	Quantitation Limit (mg/mL)	Purity (%)
*p*-coumaric acid (4)	y = 4 × 10^7^×−14.4	0.9924	0.000000–0.000008	1.10 × 10^−7^	3.32 × 10^−7^	99
4-methoxycinnamic acid (5)	y = 8 × 10^7^× + 14.5	0.9835	0.000000–0.000008	5.48 × 10^−7^	1.66 × 10^−7^	96

**Table 3 molecules-27-02359-t003:** Total phenolic and flavonoid content and DPPH scavenging activity of *Bambusa beecheyana* extracts using different extraction methods and solvent.

Extracts	TPC (mg GAE/g)	TFC (mg QE/g)	DPPH (IC_50_ µg/mL)
Cold maceration			
BBER	44.50 ± 0.03 ^a,b^	28.22 ± 0.03 ^1^	95.93 ± 0.02 ^I^
BBMR	60.15 ± 0.03 ^a^	27.73 ± 0.05 ^1^	63.32 ± 0.04 ^I^
BBHR	40.30 ± 0.02 ^b^	12.38 ± 0.04 ^2^	1931.38 ± 0.01 ^II^
Soxhlet			
BBES	97.25 ± 0.02 ^a^	40.00 ± 0.01 ^1^	87.12 ± 0.03 ^I^
BBMS	107.65 ± 0.01 ^a^	48.89 ± 0.05 ^2^	40.43 ± 0.02 ^I^
BBHS	68.95 ± 0.03 ^b^	22.39 ± 0.03 ^3^	1670.71 ± 0.03 ^II^
Ultrasonic-assisted			
BBEU20	42.65 ± 0.04 ^a,d^	25.40 ± 0.02 ^1^	573.56 ± 0.02 ^I,II^
BBEU40	55.35 ± 0.01 ^a,b^	34.45 ± 0.04 ^2^	557.20 ± 0.03 ^I,II^
BBEU60	69.60 ± 0.03 ^b,c^	36.07 ± 0.02 ^2^	463.54 ± 0.02 ^I,II^
BBMU20	58.30 ± 0.01 ^a,b^	34.46 ± 0.03 ^2^	235.71 ± 0.02 ^I^
BBMU40	85.35 ± 0.01 ^c^	35.43 ± 0.01 ^2^	45.01 ± 0.03 ^I^
BBMU60	81.85 ± 0.01 ^c^	37.20 ± 0.01 ^2^	94.27 ± 0.02 ^I^
BBHU20	42.40 ± 0.04 ^a^	25.32 ± 0.03 ^1^	982.13 ± 0.01 ^II,III^
BBHU40	45.79 ± 0.07 ^a,d^	38.32 ± 0.01 ^2^	1418.35 ± 0.03 ^III^
BBHU60	27.89 ± 0.03 ^d^	17.01 ± 0.01 ^3^	1279.95 ± 0.03 ^III^
Positive control			
Ascorbic acid	-	-	45.50 ± 0.01

The experiment was done in triplicate and the data expressed as mean ± SEM, with n=3. Data within rows with a common superscript alphabet are not significantly different from others chemical at TPC (*p <* 0.05), superscript number are not significantly different from others chemical at TFC (*p <* 0.05) and superscript roman numerals are not significantly different from others chemical at DPPH (*p <* 0.05) (two-way ANOVA, followed by Tukey’s test).

**Table 4 molecules-27-02359-t004:** List of selected cinnamic acid derivatives tentatively found in BBMR, BBMS, and BBMU40.

No.	RT (min)	Experimental *m*/*z*	Calculated *m*/*z*	Error (ppm)	Molecular Formula	MS/MS Product Ions	Tentative Identification	Ref.
1	1.036	193.0624	194.0697	−2.27	C_10_H_10_O_4_	146.0504, 134.8932, 106.0429	Ferulic acid	[30]
2	1.801	147.0585	148.0658	−2.24	C_9_H_8_O_2_	134.0152, 106.0415	Cinnamic acid	[31]
3	2.605	137.0179	138.0251	0.18	C_7_H_6_O_3_	93.0263	2-hydroxybenzoic acid	[30]
4	3.295	163.0325	164.0397	−2.77	C_9_H_8_O_3_	146.0453, 134.8934, 106.029	*p*-Coumaric acid	[30]
5	5.090	177.0480	178.0552	−1.72	C_10_H_10_O_3_	146.0464, 134.0172, 106.0429	4-methoxycinnamic acid	[32]

**Table 5 molecules-27-02359-t005:** Comparison of TPC, TFC, and DPPH scavenging activity of *Bambusa beecheyana* and various species of bamboos.

Raw Materials	Extraction Methods	Results									Ref.
		TPC (mg GAE/g)			TFC (mg QE/g)			DPPH (IC_50_ µg/mL)			
		Et. *	Me. *	Wt. *	Et. *	Me. *	Wt. *	Et. *	Me. *	Wt. *	
*Bambusa beecheyana*	maceration	44.50 ± 0.03	60.15 ± 0.03	40.30 ± 0.02	28.22 ± 0.03	27.73 ± 0.05	12.38 ± 0.04	95.93 ± 0.02	63.32 ± 0.04	1931.38 ± 0.01	This study
	Soxhlet	97.25 ± 0.02	107.65 ± 0.01	68.95 ± 0.03	40.00 ± 0.01	48.89 ± 0.05	22.39 ± 0.03	87.12 ± 0.03	40.43 ± 0.02	1670.71 ± 0.03	
	ultrasonic- assisted	42.65 ± 0.04	58.3 ± 0.01	42.4 ± 0.04	25.4 ± 0.02	34.46 ± 0.03	25.32 ± 0.03	573.56 ± 0.02	235.71 ± 0.02	982.13 ± 0.01	
		55.35 ± 0.01	85.35 ± 0.01	45.79 ± 0.07	35.45 ± 0.04	35.43 ± 0.01	38.32 ± 0.01	557.2 ± 0.03	45.01 ± 0.03	1418.35 ± 0.03	
		69.6 ± 0.03	81.85 ± 0.01	27.89 ± 0.03	36.07 ± 0.02	37.2 ± 0.01	17.01 ± 0.01	463.54 ± 0.02	94.27 ± 0.02	1279.95 ± 0.03	
*Bambusa tulda*	maceration		126 ± 3.4	-	-	40 ± 0.2	-	-	360 ± 1.4	-	[12]
	Soxhlet	-	164 ± 3.8	-	-	68 ± 0.9	-	-	404 ± 4.3	-	
*Bambusa arundinacea*	maceration		14.6	2.79	-	6.71	2.54	-	273	964	[13]
	ultrasonic-assisted	-	647.76 ± 5.77	-	-	247.85 ± 3.79	-	-	-	-	[20]
*Bambusa vulgaris*	maceration	44 ± 0.1	-	27 ± 0.5	22 ± 0.3	-	12 ± 1	490 ± 60	-	400 ± 20	[14]
*Bambusa nutan*	maceration	-	15.35 ± 0.55	-	-	-	-	-	123.45	-	[16,18]
		-	-	180.45	-	-	-	-	-	85.81	
	Soxhlet	-	230.07	-	-	139.11	-	-	57.89	-	
*Phyllostachys bambusoides*	maceration	-	-	-	-	-	-	882.08	-	-	[41]
*Gigantochloa levis*	maceration	2500	-	-	-	-	-	86.4 ± 1.05	-	-	[42]

* Et.—ethanol extract, Me.—methanol extract, Wt.—water.

## Data Availability

Not applicable.

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
