# Peer review of "Effects of Extraction Methods on Phenolic Content in the Young Bamboo Culm Extracts of Bambusa beecheyana Munro"

_molecules, 2022, doi:10.3390/molecules27072359_

Round 1

Reviewer 1 Report

In this study, cold impregnation, soxhlet and ultrasonic assisted extraction techniques were used to extract Bambusa beecheyana Munro, and the total phenol content, total flavonoids content and free radical scavenging activity of Phyllostachys pubescens were determined. The identification and quantification of phenolic components were completed. This is an interesting work. However, some revision is required and the following points should be addressed: 1. 4.2 Extraction method of Bmbusa beecheyana Culm. Why powdered of young bamboo culm with different weights are used in the different extraction methods? 2. 4.2 Extraction method of Bmbusa beecheyana Culm. Please explain why the experiment is designed in this way? How to determine the extraction temperature and soaking time? If you refer to other people's articles, please list them. 3. It can be seen from table 1 that the contents of coumaric acid and 4-methoxycinnamic acid, especially coumaric acid, did not change significantly under different extraction methods. Please explain whether the subsequent separation, purification and content determination of coumaric acid and 4-methoxycinnamic acid are redundant? 4. Antioxidant capacity generally has a great correlation with the content of polyphenols and flavonoids. This study only measured their content, and did not discuss the correlation between them in depth. Please consider whether to supplement this content. 5. 2.2 HPLC chromatograms of other extraction methods can be considered to be provided in supplementary materials. 6. The manuscript suffers grammatical mistakes which should revised and corrected. The language could be better.

Reviewer 2 Report

It was a challenge to review the article "Effects of Extraction Methods on Phenolics Content in the young bamboo culm extracts of Bambusa beecheyana Munro" by Mohd. Izuddin Nuzul et al..

This manuscript is not enough for publication. It requires several improvements. I annex the revised manuscript with some suggestions and comments for the authors.  

Reviewer 3 Report

Reviewer report

There is a huge amount of literature on the optimization and comparison of extraction methods for the isolation of bioactive compounds from plants. A significant part of the literature involves phenolics and other antioxidants.

To my opinion this work offers limited new knowledge to the audience of a general journal such as Molecules (well established extraction strategies are discussed in a specific application). My suggestion is towards submitting this work to a more specialized journal on Phytochemistry.

Either way, extensive lingustic revision is required prior to further processing.   

Author Response

Thank you for your suggestion. Sorry for the English language mistakes. We have done the amendment on this. 

Reviewer 4 Report

The manuscript presented the study of phenolic compounds extraction and identification in Bambusa beecheyana. Some improvements are suggested to increase the quality of the manuscript:

Introduction:

Line 92: Moreover, in depth bioactive compounds were characterized and isolated using HPLC, GC/MS,  UV-Vis, FTIR and NMR. 

Question:  Do you have any results from GC/MS, FTIR, and NMR? I can not see them in the manuscript.

Result:

1- The authors presented sets of data on different extraction methods and compared the yields. Different extraction methods were performed with different (solid:liquid) ratio. How can you make a comparison regarding the yield of extraction?

2- Figure one can be bigger and more clear by deleting the unnecessary small peaks. Also, it would be good to add chromatogram of standards.

3- Line 224: There are just 2 peaks in figure 1. Please check the sentence.

4- There are two (figure 2) and two (table 5) without any table 4 in the manuscript.  Please check the figures and tables number. 

Materials and methods:

It would be good to put section (4.11 comparison with other studies) in the discussion part.

Round 2

Reviewer 2 Report

The latest version of the manuscript has improved its quality. I thank the authors for all the effort, rigour, and dedication in their responses. I have no objection to the publication of this paper in the "Molecules" in its current form.

Reviewer 3 Report

The revised version is ok!